# AI-Enabled Traffic Control Prioritization in Software-Defined IoT Networks for Smart Agriculture

**DOI:** 10.3390/s23198218

**Published:** 2023-10-02

**Authors:** Fahad Masood, Wajid Ullah Khan, Sana Ullah Jan, Jawad Ahmad

**Affiliations:** 1Department of Electronics, Quaid i Azam University, Islamabad 45320, Pakistan; 2Department of Computing, Abasyn University, Peshawar 25000, Pakistan; arbabwajid.ullah@abasyn.edu.pk; 3School of Computing, Engineering and the Built Environment, Edinburgh Napier University, Edinburgh EH10 5DT, UK; s.jan@napier.ac.uk (S.U.J.); j.ahmad@napier.ac.uk (J.A.)

**Keywords:** SDN, IoT, emergency/critical data, smart agriculture system, machine learning, reinforcement learning

## Abstract

Smart agricultural systems have received a great deal of interest in recent years because of their potential for improving the efficiency and productivity of farming practices. These systems gather and analyze environmental data such as temperature, soil moisture, humidity, etc., using sensor networks and Internet of Things (IoT) devices. This information can then be utilized to improve crop growth, identify plant illnesses, and minimize water usage. However, dealing with data complexity and dynamism can be difficult when using traditional processing methods. As a solution to this, we offer a novel framework that combines Machine Learning (ML) with a Reinforcement Learning (RL) algorithm to optimize traffic routing inside Software-Defined Networks (SDN) through traffic classifications. ML models such as Logistic Regression (LR), Random Forest (RF), k-nearest Neighbours (KNN), Support Vector Machines (SVM), Naive Bayes (NB), and Decision Trees (DT) are used to categorize data traffic into emergency, normal, and on-demand. The basic version of RL, i.e., the Q-learning (QL) algorithm, is utilized alongside the SDN paradigm to optimize routing based on traffic classes. It is worth mentioning that RF and DT outperform the other ML models in terms of accuracy. Our results illustrate the importance of the suggested technique in optimizing traffic routing in SDN environments. Integrating ML-based data classification with the QL method improves resource allocation, reduces latency, and improves the delivery of emergency traffic. The versatility of SDN facilitates the adaption of routing algorithms depending on real-time changes in network circumstances and traffic characteristics.

## 1. Introduction

Software-defined networks (SDNs) have evolved as potential tools for managing and controlling modern network infrastructures [1]. The ability of SDNs to separate the control and data layers allows for flexible network setups, effective resource management, and granular traffic control, as shown in Figure 1. SDNs have found use in a number of industries, including transportation, healthcare, smart cities, and agriculture. Modern communication networks have undergone substantial changes due to the rise of Software-Defined Networking (SDN). The administration of network resources is programmable, dynamic, and extensible thanks to SDNs’ isolating of the control plane from the data plane. This makes it possible to create intelligent and flexible network systems for various uses, including in the agricultural sector [2,3].

Agriculture is one of the most important areas of the global economy; with a rising population comes an inevitable increase in food consumption. Smart agriculture, which combines cutting-edge technologies with traditional farming methods, has the ability to transform how we create food and handle agricultural resources [4,5,6,7,8,9]. SDN-based solutions can promote smart agriculture by offering a dynamic and clever network architecture in support of a broad variety of applications, such as precision agriculture, weather tracking, and livestock management [10]. In modern agriculture, sensors play a critical role in collecting information about various factors such as water, soil, climate, etc., as shown in Figure 2. Analysis can be carried out with the help of data obtained from different sensors to identify and improve the current situation of crop production [11]. The progressive variability of acquired measurements is an essential aspect of the agricultural domain that requires significant attention. The scale of agricultural production and industrial weakness can be significantly changed through modern agriculture. Moreover, it has an imperative role in agriculture development, as well as in the realization of a healthy society [12].

Traditional data processing techniques in smart agricultural systems face numerous severe hurdles when dealing with the complexities of data management. The massive amounts of data created by sensors and devices can overwhelm traditional workflows, resulting in slow analysis and decision-making processes. This is worsened by the wide range of data types, which necessitate specialized processing capabilities. The importance of real-time monitoring heightens the requirement of quick data handling, which traditional systems fail to satisfy [13,14,15,16]. In this context, data can be managed and analyzed with the help of event-based data analysis methods. Event-based data analysis can assist in locating patterns, trends, and anomalies in the data that can then be used to enhance agricultural decision-making by identifying and analyzing events in real time. Thanks to its ability to autonomously identify trends and correlations in massive datasets, machine learning (ML) is a potent instrument for event-based data analysis. Real-time tracking of agricultural circumstances is made possible by the use of software-defined networks and ML, offering useful information for improved decision-making.

In this article, we suggest an event-based traffic control prioritization framework for smart agriculture that makes use of ML techniques. The optimized traffic routing begins with the categorization of incoming data using various ML techniques, such as Logistic Regression (LR), Random Forest (RF), k-nearest Neighbours (KNN), Support Vector Machine (SVM), Naive Bayes (NB), and Decision Tree (DT). These algorithms analyze characteristics such as as humidity, temperature, wind speed, leaf moisture, soil temperature, and soil moisture to classify data traffic into emergency, normal, and on-demand categories. Following classification, Q-learning, a fundamental reinforcement learning (RL) algorithm, manages dynamic routing in the SDN configuration. SDN allows the system to dynamically handle network traffic and resources based on the requirements of the application. The SDN design enables a centralized control layer that makes network management and configuration simpler. It keeps a Q-table with the projected cumulative reward for certain actions in specific data class conditions. QL converges to optimal Q-values through research and application, suggesting the appropriate actions for each data class. The SDN controller examines the Q-table during traffic routing to make decisions according to the previous routing performance of the relevant data class. This integrated system modifies routing pathways based on data traffic type, ensuring effective and fast decisions regarding routing in the SDN environment. The combination of ML’s categorization expertise with QL’s flexible decision-making results in a versatile system that optimizes data routing while allowing for real-time adaptation.

The use of event-based traffic analysis serves to reduce data duplication and improve data processing efficiency, while ML algorithms help to categorize the data into various groups, such as normal or emergency conditions. The proposed technique employs QL to dynamically alter routing decisions in accordance with real-time data classes and past performance. This advancement enables optimal resource utilization and reduced latency. Furthermore, the use of ML-based data categorization enables class-based routing, which improves network intelligence and customizes routing decisions. The innovative component is the interaction of ML classification with QL adaptive routing. This technique provides real-time optimization, comprehensive traffic management, and empirical confirmation in a simulated environment by overcoming the gap between efficient categorization and dynamic routing. The implementation of this research in smart agriculture has the potential to increase agricultural yields, cut water use, and improve resource management. Moreover, the execution of the system is made flexible and scalable through the use of a software-defined IoT network.

The rest of this paper is organized as follows: Section 2 offers a comprehensive overview of related work in the fields of SDN-based smart agriculture and event-based traffic analysis; Section 3 covers the suggested methodology in depth, including the dataset used for experimentation; Section 4 contains the testing results, along with a comparison of the various ML methods used; finally, Section 5 concludes the paper by highlighting upcoming research paths.

## 2. Literature Review

ML methods are increasingly being used nowadays. These methods are thought to be superior to conventional algorithms, especially when handling and analyzing big data. Researchers are focusing on the application of such techniques in the field of networks. ML has found varied uses in the area of SDN, including traffic engineering [17,18], resource management [19,20], intrusion detection systems [21,22], and other security objectives [23,24]. In this regard, Akyildiz et al. [25] presented the state of the art for traffic engineering in SDN/OpenFlow networks. Mijumbi et al. [26] used ML to adjust virtual networks and control resources in virtualized networks using a control plane. As a result, the significance of ML in SDN has increased of late due to its numerous applications. SDN’s architectural reasoning works better with ML algorithms than with conventional algorithms. In particular, numerous research findings have used SDN and ML methods in combination to optimize routing. Furthermore, ML is considered a crucial technology development for 6G and beyond [27].

Etengu et al. thoroughly evaluated AI-assisted networks for load balancing and green routing. Their analysis centered on a pragmatic strategy, hybrid SDN, which is typically utilized for smooth migration from legacy systems [28]. A collection of challenges and prospective research paths were discussed and a particular framework for handling them was proposed. Qian et al. provided a succinct overview of a variety of use cases in communication networks that rely on reinforcement learning, such as network caching and task sharing [29], although their review barely mentioned the connection between routing apps and SDN. Mammeri et al. thoroughly examined reinforcement learning approaches for routing for SDN-based networks as well as for other kinds of networks, providing a very excellent overview of the evolution of this particular ML technique and its implementation in communication networks [30].

Jamshidi et al. classified ML-based apps into six networking categories: network security, traffic prediction, cloud services, domain name system, application identity, and QoS. They then selected the best ML algorithms and raw datasets for each of these groups [31]. This approach highlights the key problems and outcomes of these raw data and ML techniques. Zhang et al. showed various uses of ML in resource allocation and routing in optical networks, though with no particular emphasis on SDN-enabled networks [32]. Boutaba et al. surveyed ML research possibilities and evolution in the area of networking [33], providing a short summary of ML techniques in routing, anomaly detection, traffic categorization, fault management, QoS/QoE, and intruder detection. The engineering techniques, approaches, and methods for data collection in network traffic were discussed as well, and they emphasized the value of online learning, safe learning assistance, and system architectures that make it simple to use ML. Xie et al. provided a thorough description of ML techniques and of the design and operation of SDNs [34] in terms of QoE/QoS, optimization, resource management, security, and traffic categorization, various ML algorithm types. Zhao et al. reviewed the various networking applications that profit from the integration of SDN and ML, including a brief discussion of routing optimization [35].

Tamizhselvan and Vijayalakshmi discussed an SDN-based solution named “SDN-MCHO” designed to improve reliable device routing in IoT contexts, especially for smart surveillance applications [36]. Their key focus was on leveraging Software-Defined Networking (SDN) to optimize routing decisions in IoT networks. Sharma et al. provided a method called “FCS-fuzzy net” that handles CH selection as well as routing for weed categorization in IoT contexts [37]. Their study emphasized the use of fuzzy logic, and relied on a framework known as MapReduce for effective data processing and routing in IoT networks. Fuzzy logic was used to improve the decision-making process for picking cluster heads, which play an important role in organizing data routing in IoT systems, with a specific application focus on weed categorization. Naeem et al. used SDN to provide a unique solution to energy-efficient routing optimization in the Industrial Internet of Things (IIoT) [38]. Their major goal was to improve energy efficiency in IIoT networks by optimizing routing decisions using SDN capabilities. In light of the vital role of energy management in IIoT, their study is particularly pertinent with respect to industrial applications, where effective routing can result in considerable energy savings and enhanced network efficiency.

## 3. Materials and Methods

We consider the band communication model with the SDN controller having reactive flow installation mode and open flow enabling the switch, as shown in Figure 3. The employment of a controller in this study, particularly in the context of Software-Defined Networking (SDN), provides numerous major advantages. First of all, it centralizes network administration, effective decision-making, and allocation of resources. Second, the controller allows for dynamic routing adaption based on data classification, guaranteeing that data packets are routed appropriately depending on their priority and features. The network’s flexibility improves its efficiency and responsiveness. Furthermore, the controller optimizes resource allocation, resulting in better resource utilization and lower congestion, and plays an important role in reducing latency for vital data types and improving data delivery dependability. The proposed approach contains processes with an ultimate goal of assisting the agricultural sector through a merged SDN paradigm and AI modeling for normal, emergency, and data on demand operation. The first step is to collect data from field sensors for various features. Data acquired from the sensors are preprocessed before being sent to the ML Classification Layer. After collection, the data must be cleaned and transformed into a format appropriate for machine learning models. After preprocessing, the data are divided into training and testing sets. The training set is then used to train an ML model. The model is finally tested on the testing dataset to evaluate the accuracy. After training and testing, the model may be used to categorize traffic as normal, emergency, or data on demand. The categorized data traffic is then passed to the SDN control layer, where the controller uses the Q-learning algorithm to dynamically route traffic flow depending on the data classifications. The Q-learning algorithm adapts to the present state of the network and performs actions to optimize a reward signal, such as prioritization, efficient routing, and bandwidth allocation. The Q-values are modified depending on learned rewards, and the SDN controller alters traffic flow control mechanisms appropriately. The detailed workflow process is discussed below.

### 3.1. Data Collection

A network of sensors strategically positioned in the city of Peshawar, Pakistan was used to collect data. These sensors were meticulously placed to account for a variety of determining factors, such as particular crop requirements, varying environmental conditions, and the optimum range of each sensor type. To avoid interference, the sensors were equally spaced, and were placed in accordance with the specific data requirements. Locations for sensors were chosen based on their accessibility to installation sites and their cost effectiveness. Data for six different crop types (wheat, mint, coriander, radish, turnip, and carrot) were collected over a four-month period from mid-October 2022 to mid-February 2023. The chosen crops exhibit a variety of preferences with respect to climatic conditions, soil properties, and other variations. Wheat grows best in loamy soils that are well-drained and moderate. On the other hand, mint and coriander prefer somewhat warmer and more humid settings, necessitating well-drained soils rich in compost. Root vegetables, including turnip, carrot, and radish, are tolerant to a wide range of climate conditions and soil types, generally preferring loose and well-drained soils.

For full data coverage, a total of twenty-five sensors were deployed within a range of 1000–3000 m over the agricultural region to measure humidity, temperature, wind speed, leaf moisture, soil temperature, and soil moisture. The sensors were placed for comprehensive environmental data gathering while considering the field’s size and the research objectives. Three sensors were positioned for temperature and humidity monitoring: one dedicated to wheat, another to mint and coriander collectively, and a third to carrot, radish, and turnip collectively. Similar combinations were employed for the leaf moisture and wind speed sensors. Each of these sensor groups was tailored to the specific requirements of the crops within their collective category. For critical factors such as soil moisture and soil temperature, additional sensors were dedicated: six to soil temperature and seven for soil moisture for each crop, with two sensors out of these latter seven dedicated to wheat due to the larger crop area.

The specialized sensors were carefully selected to guarantee precise and complete monitoring of critical environmental parameters: capacitive humidity sensors (DHT22, TZT, China)for humidity levels in the air, resistance temperature detectors (RTD PT100, MKYD, China) for temperature measurement, anemometers (UT363BT, UNI T, China) for measuring wind speed and providing information about weather conditions, leaf wetness sensors (LWS-31, LIYUAN, China) for monitoring the leaf moisture, thermocouples (MAX6675, Thermocouple Module + K type Sensor, TZT, China) to measure soil temperature, and volumetric soil moisture sensors (Smart Electronics Soil Moisture Hygrometer Detection Humidity Sensor, STLXY, China) for the soil moisture content. The sensors were connected to an Arduino microcontroller (UNO-R3, TZT, China) for data collection, and the data collected by the sensors was transmitted via Wi-Fi to a central hub for further processing. The data were then preprocessed to remove any unusual or unnecessary data before being translated into a suitable format for the machine-learning models. The preprocessed data were divided into training and testing sets in order to train and assess the performance of several different machine learning models. The data used in this research can be accessed at the following repository: https://github.com/researchcsaup/IoTs.git (accessed on 5 August 2023).

### 3.2. ML Models

ML is a prominent application for artificial intelligence, as it automates the system and enables it to learn and develop. The ML learning process begins with the observation of data through cases or observations. These data contain patterns that, when found, can support more accurate predictions. Using the test dataset, the six alternative ML models discussed below were employed to train the classifiers, then their classification performance was assessed.

#### 3.2.1. Logistic Regression

Logistic regression is a binary classification method that employs a logistic function to describe the likelihood of a binary response variable [39]. The logistic function on which the algorithm is built converts any input to a number between 0 and 1. This approach is employed to estimate the likelihood of a specific occurrence.

#### 3.2.2. Decision Trees

Decision trees are a form of supervised learning algorithm commonly used for classification issues [40]. They utilize a model of choices and potential outcomes that resembles a tree. Decision trees operate by recursively dividing the data into smaller groups, which they do by selecting the feature that best divides the data according to certain parameters, such as the information gain or the Gini index.

#### 3.2.3. Random Forest

Random forests are an expansion of decision trees founded on the concept of generating numerous decision trees, each with a random portion of the data, then combining their findings to enhance overall performance [41]. Random forests are frequently used for categorization issues, especially when working with higher-dimensional data.

#### 3.2.4. Naive Bayes

Naive Bayes is a class of probabilistic algorithm that uses Bayes’ theorem to forecast the likelihood of an occurrence happening [42]. In order to determine the likelihood of a class given a collection of characteristics, the Gaussian Naive Bayes algorithm relies on the presumption of the features being normally distributed.

#### 3.2.5. Support Vector Machine

Support vector machines are a form of supervised learning algorithm used to solve classification and regression issues [43]. The SVM algorithm divides data into groups by locating a hyperplane in a high-dimensional region. The SVM approach is especially helpful when working with data that are not linearly separable, as it uses a kernel functions to convert the data to a higher-dimensional space where they can be divided.

#### 3.2.6. K-Nearest Neighbros

K-nearest neighbors is a simple classification method that works by locating the k nearest data points in the training set with respect to a particular test point and then predicting the test point’s class based on the majority class of its k-nearest neighbors [44]. KNN is a non-parametric method, which means that it does not make any assumptions about the distribution of the data.

### 3.3. Bootstrapping

In the bootstrapping technique, Hello, Feature-Request, and Feature-Reply messages are exchanged via the Open Flow protocol when the network is switched on. This takes place between the controller and the switch to obtain the network’s global view of the controller. The Feature-Request message is periodically generated by the controller through Hello messages to obtain the switch features.

The switch sends a Feature-Reply message to the controller after receiving the Feature-Request message; the controller obtains the switch’s capabilities in this process. As discussed earlier, reactive flow installation mode is assumed. Unlike traditional forwarding devices, the switches have no awareness in this mode when the network is configured for routing and first starts running. When the data packet (PackectIn) of a flow arrives at the switch, the switch immediately looks for the matching entry in its forwarding table; if the matching flow entry is found, it forwards the said data packet using the corresponding action in the flow table entry. Otherwise, the switch asks the controller to compute the action for the flow.

The controller checks the network reachability rule for the flow among source and destination IPs; whether it is allowed or denied is specified at the central controller next to the receiving request from the switch. If the flow is denied, the controller installs the drop action at the switches; otherwise, it computes the primary path, for which various approaches can be used. When a switch is connected to a controller, the controller periodically sends commands to the devices through the link layer protocol of the discovery broadcast domain’s discovery protocol via all interfaces of the switches. A discovery packet contains the data path identification of the sending host along with the interface information that generates the packet for the destination end. Occupied sets of destination device MAC addresses and the Ethernet type are differentiated from other kinds of packets in the network by the controller based on the link layer discovery protocol (LLDP). The LLDP is widely used for discovery of direct links to the next hop in a network (for instance, among two switches), while the broadcast domain discovery protocol is widely used for device discovery in the same domain.

### 3.4. Controller Event Composition in Graph Theory

Event-driven application behavior has the composition of all SDN bindings (controller, server, forwarding devices); in this paper, we use graph theory to support our proposed approach. Applications use a hash object (dictionary) to store nodes, attributes, link properties, communication channel attributes over SSL, and state-of-the-art algorithmic path computation optimization. Graph composition can be inducted through the following procedure in the POX controller: (1)G=(V,E),
where “*V*” represents the end nodes and forwarding devices in the network and “*E*” is the set of the edges of devices and end nodes in the network.

More specifically, a network translated or composed in the format of the graph encompasses the following attributes:

V= (MAC, IP, sensor connected, local sensor controller connected, time in network, event on addition and deletion).

E= (edges among the nodes (sensors), edges among the controller in the network, edges from switches to the sensor conveyer or controller, time stamp of joining the SDN network, time of leaving the network).

### 3.5. Alternative Path Computation in Case of Critical Sensor Traffic

In this process, our control application is intended to find a path with intermediate forwarding devices that are not included in the controller dictionary record of the installed flow rule. Critical traffic requires alternate paths to enable speedy control of traffic by the controller specification and receiving at the server.

*OFPFC_ADD* is an Open Flow command used for the flow rule installation in the flow table of the switch. The match fields encapsulated in the flow-adding command are first compared for a corresponding flow rule entry in the switch. Matching objects are matched: *nw_proto* (application layer protocol), *match.dl_type* (opcode of IPV4, ARP), *match.nw_src* (source IP address) and *match.nw_dst* (destination IP address). The timeout and the priority values are specified by the controller, along with the flow rule installation command.

### 3.6. Routing Algorithm

Reinforcement Learning and Software-Defined Networking for Intelligent Routing (RSIR) introduces a knowledge plane and identifies a routing algorithm using Reinforcement Learning (RL) that takes link state information into consideration when exploring, learning, and exploiting potential paths during intelligent routing regardless of any dynamic traffic transitions. This algorithm makes use of the environment’s interaction, the intelligence offered by RL, and the global perspective for managing the network provided by SDN. It determines and implements optimum routes in the routing table of the data plane switches in advance.

The RL agent defines the flow pathways using the Q-learning approach. Q-learning is a method without models that does not require prior knowledge of the reward earned by performing a given action in a specified situation [30]. The flow of the QL-based routing is shown in Figure 4.

### 3.7. ML Performance Metrics

The performance metrics used for evaluation were the accuracy, precision, recall, and F1-score, using the following equations.
(2)$$Accuracy=\frac{TP+TN}{TP+TN+FP+FN}$$
(3)$$Precision=\frac{TP}{TP+FP}$$
(4)$$Recall=\frac{TP}{TP+FN}$$
(5)$$F1\text{-}Score=\frac{2*Precision*Recall}{Precision+Recall}$$

### 3.8. Box Plot

A box plot is a statistical graphic that shows the distribution of a dataset in the form of the median, interquartile range (IQR), and range of the data [45]. Box plots are helpful for assessing a dataset’s distribution, skewness, and probable outliers. The middle 50% of the data is indicated by the box itself, the lowest or first quartile (Q1) represents the 25th percentile, and the highest or third quartile (Q3) indicates the 75th percentile. The line inside the box shows the median value of the data. The IQR is the difference between Q3 and Q1. There are two whisker lines that extend from the outside of the box; one extends from the minimum to the lower quartile and the second from the upper quartile to the maximum. The outlier values are indicated by the small circles on top of the top whisker and at the bottom of the bottom whisker. An outlier in the data is a very high or extremely low value. The boxplot’s top whisker reflects the greatest value in the data that is not an outlier. The outlier values can be obtained as **outlier** > Q3 +(3×IQR) and **outlier** < Q1 −(3×IQR).

### 3.9. Performance Evaluation

Mininet 2.2 EEL was used as a network simulator, as it provides good ease of use for the user and offers formation and easy setting of SDN elements along with sharing, customization, and testing of the SDN network’s performance. It includes forwarding device switches, end hosts, links, and interfaces for controller interoperability. Further, it provides a separate virtual environment for executing various applications for each host. For control functions, the Pox controller event-driven approach was used. Our network simulator-based virtual scenario encompasses the resources of an HP 450 G5, Core i7-8250U, 16GB of physical memory, and Linux distribution (Ubuntu) operating system.

The topology of our network consists of 25 sensors with the same geolocation for measuring humidity level, temperature, wind speed, leaf moisture, soil temperature, and soil moisture. More precisely, each sensor contributor has an IP address. The topology consists of ten Open Flow-Enabled switches which are commanded by the Pox controller for reachability specification and installation of flow rules. These forwarding devices act upon the received commands instantaneously.

## 4. Results and Discussion

The results of the proposed number of sensor communications are discussed for various parameters i.e., sensor nodes, protocol, controller, simulation time, packet time, traffic, and total calculation time. The parameters used for the simulations are shown in Table 1.

Various evaluation metrics, such as accuracy, precision, recall, and F1 score, provided in Equations (Equation 2)–(Equation 5), were used to evaluation the models’ performance, as mentioned in Table 2. Notably, the RF and DT models outperform the other ML models in terms of accuracy. A graphical presentation of the results for all the ML models is shown in Figure 5. Overall, the RF and DT models perform well across all measures, indicating their ability to accurately categorize data into the required categories. The performance of the KNN and SVM models is low compared to that of the LR and NB models, which in turn perform less effectively than the RF and DT models.

The box plot results for all the features are presented in Table 3 and Figure 6 and Figure 7. A detailed analysis of the results for each feature is provided below.


**Humidity Level:**


The mean humidity level is 28, with 75% of the data being less than 31 and 25% being less than 26. The maximum humidity level is 52, the minimum humidity level is 24, and there are 106 outliers. The IQR is 5, with 38.5 being the highest outlier and 18.5 being the lowest outlier.


**Temperature:**


The mean temperature is 20, with 75% of the data being less than 23 and 25% being less than 18. The maximum temperature is 32, the minimum temperature is 8, and there are twelve outliers. The IQR is 5, with 30.5 being the highest outlier and 18.5 being the lowest outlier.


**Wind Speed:**


The mean wind speed is 9, with 75% of the data being less than 10 and 25% being less than 8. The maximum wind speed is 20, the minimum wind speed is 7, and there are 98 outliers. The IQR is 2, with 13 being the highest outlier and 5 being the lowest outlier.


**Leaf Moisture:**


The mean leaf moisture is 86, with 75% of the data being less than 92 and 25% being less than 84. The maximum leaf moisture is 97, the minimum leaf moisture is 66, and there are no outliers. The IQR is 8, with 104 being the highest outlier and 72 being the lowest outlier.


**Soil Temperature:**


The mean soil temperature is 17, with 75% of the data being less than 19 and 25% being less than 15. The maximum soil temperature is 24, the minimum soil temperature is 12, and there are no outliers. The IQR is 4, with 25 being the highest outlier and 9 being the lowest outlier.


**Soil Moisture:**


The mean soil moisture is 13, with 75% of the data being less than 14 and 25% being less than 12. The maximum soil moisture is 15, the minimum soil moisture is 6, and there are no outliers. The IQR is 2, with 17 being the highest outlier and 9 being the lowest outlier.

Figure 8 presents a comparison of traffic types based on their average delay characteristics. Because of the high frequency of data transfers in this category, “normal traffic” has a significantly larger average latency in this context. The increased frequency of regular data packet transmission causes network congestion, which contributes to an increase in the average delay time. When “critical packet transmission” is considered, the scenario changes, resulting in a path with less traffic. As this path encounters less congestion, there are shorter wait times in this scenario compared to the usual traffic category. This difference in latency can be related to differences in traffic flow and packet reception frequencies, where normal traffic consists of a continuous flow of data packets; critical packet transmission comprises less traffic, and as a result, less congestion and shorter delay periods.

Figure 9 shows the average packet loss encountered by the three data categories of emergency, normal, and on-demand. Notably, the mean delays for emergency traffic are consistently shorter than those for regular and on-demand traffic. The reason for this disparity is that the RSIR algorithm is used for emergency data. RSIR prioritizes the use of shorter and less congested paths for critical data packets. As a result, critical data packets experience fewer delays and less packet loss. On the other hand, normal and on-demand traffic, which may use alternative routing algorithms, tend to face significantly longer delays, and as a result experience higher average packet loss.

Figure 10 represents the average throughput over the course of a week for the three data traffic types of emergency, on-demand, and routine. It can be noticed that critical and on-demand data traffic follow a unique pattern defined by a broader dispersion of flows over the network. When compared with normal data flow, this phenomenon results in the use of a higher number of comparatively less-used channels. Surprisingly, despite the decreased packet loss and mean delay encountered by critical and on-demand data, the mean throughput for each category appears to be lower than for normal traffic. It is worth mentioning that critical and on-demand data receive advantages such as decreased data loss and shorter delays as a result of their more effective routing. The diversification of their flows across multiple network channels may result in less optimized use of available bandwidth, resulting in reduced bandwidth. This highlights the complexities of routing schemes and their influence on network efficiency.

## 5. Conclusions

In this research, we have presented an event-based data analysis technique for smart agriculture systems using the Internet of Things (IoT) based on machine learning (ML) models such as Logistic Regression (LR), Random Forest (RF), k-Nearest Neighbours (KNN), Support Vector Machine (SVM), Naive Bayes (NB), and Decision Tree (DT). The models’ accuracy was used to evaluate their performance for features such as humidity level, temperature, soil moisture, wind speed, etc., finding acceptable levels of accuracy. The proposed approach makes use of software-defined networking (SDN) capabilities to identify and manage essential network events. A software-defined networking (SDN) controller and the Q-learning algorithm are then used to route the data traffic. Our results indicate that the proposed technique can efficiently manage the data flow. The usefulness of merging ML and SDN for intelligent routing and network performance optimization is demonstrated by this research. Having explored the benefits of utilizing advanced algorithms for data classification and routing in a software-defined IoT network, the scalability and application of this approach in larger or heterogeneous networks may be an interesting area for future research.

## Figures and Tables

**Figure 1 sensors-23-08218-f001:**
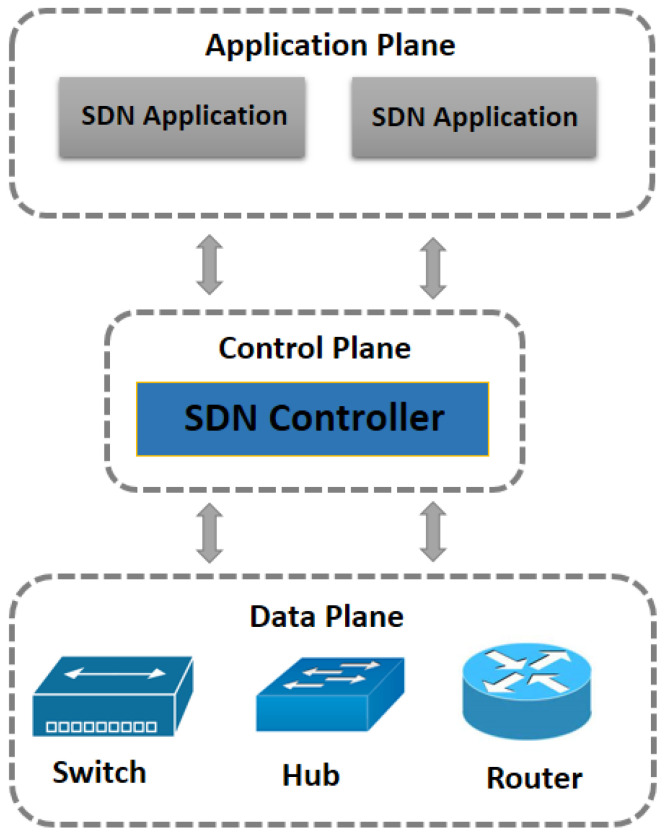
SDN architecture.

**Figure 2 sensors-23-08218-f002:**
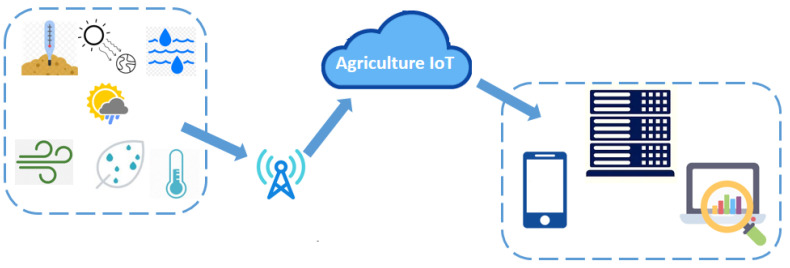
Traditional IoT scheme for smart agriculture.

**Figure 3 sensors-23-08218-f003:**
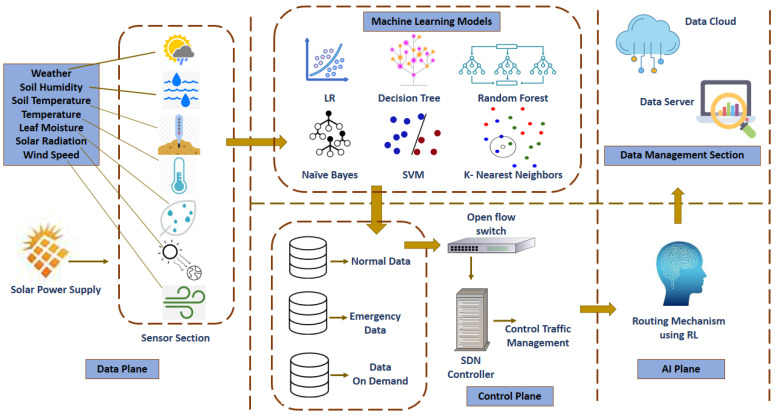
System architecture.

**Figure 4 sensors-23-08218-f004:**
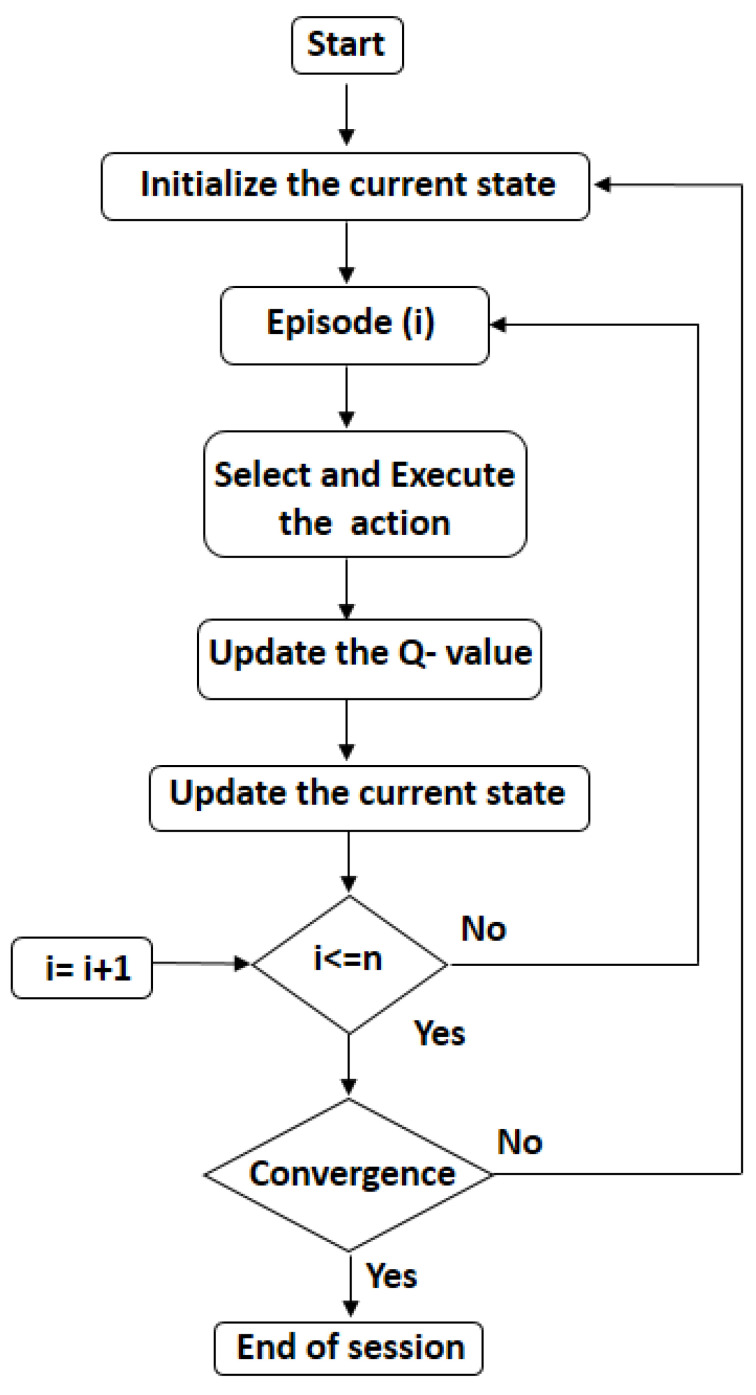
Q-Learning algorithm.

**Figure 5 sensors-23-08218-f005:**
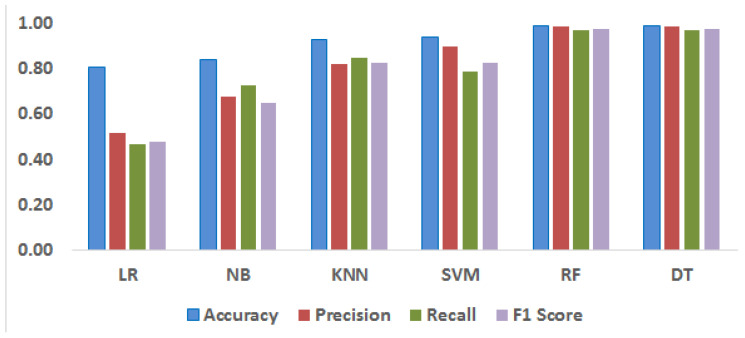
Comparative analysis of various ML models.

**Figure 6 sensors-23-08218-f006:**
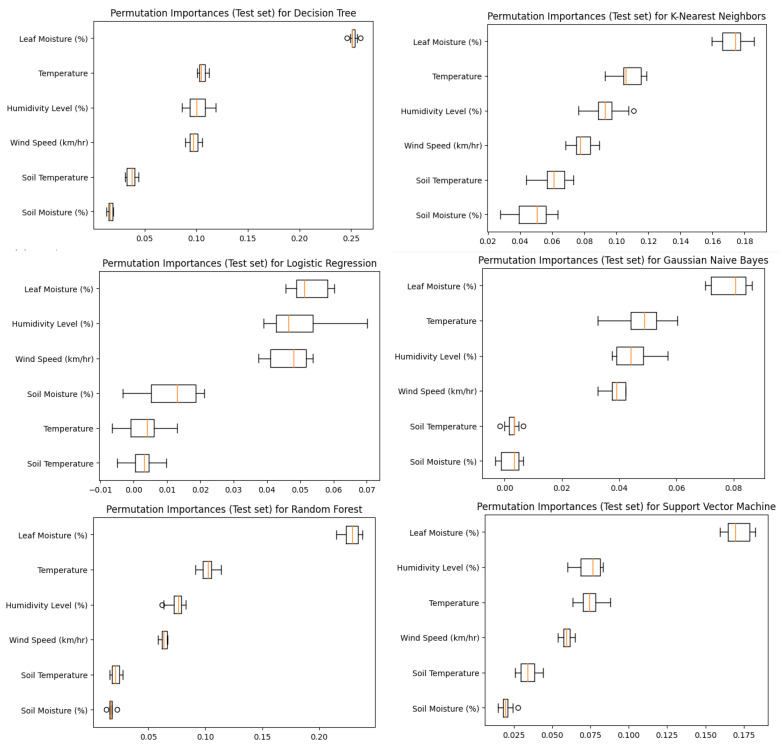
Comparative analysis of the data distribution for various parameters.

**Figure 7 sensors-23-08218-f007:**
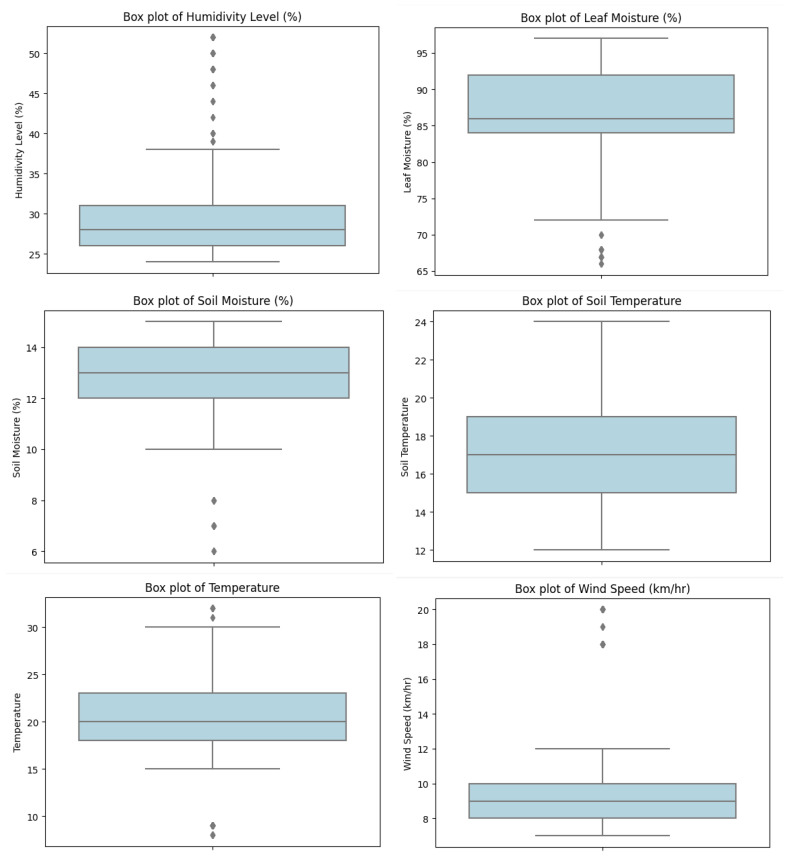
Data distribution analysis for the individual parameters.

**Figure 8 sensors-23-08218-f008:**
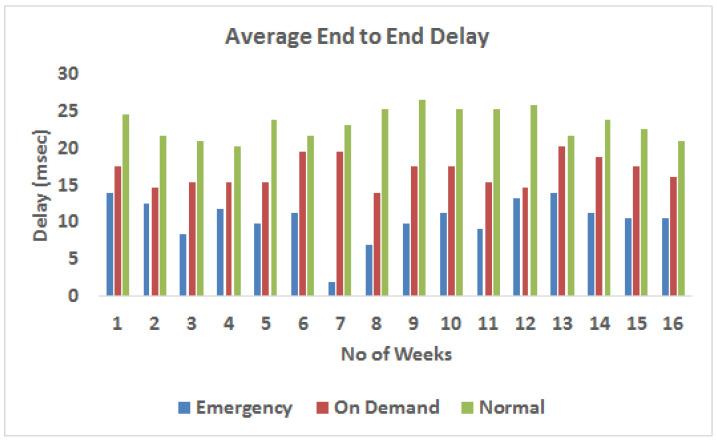
Average delay throughout the week.

**Figure 9 sensors-23-08218-f009:**
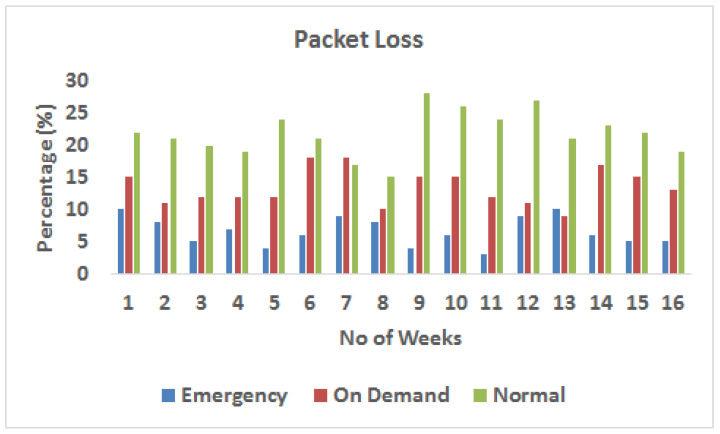
Average packet loss throughout the week.

**Figure 10 sensors-23-08218-f010:**
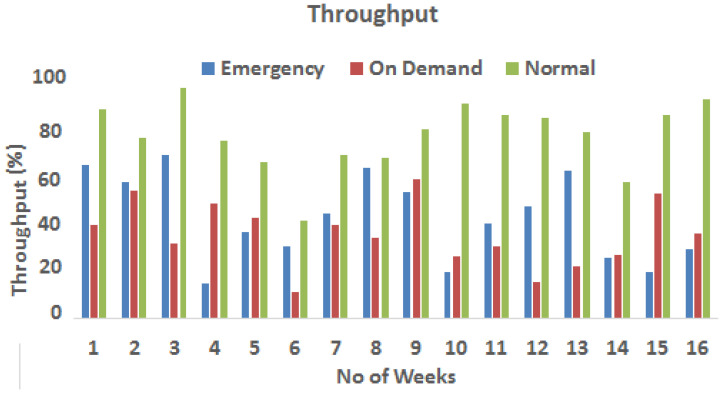
Average throughput throughout the week.

**Table 1 sensors-23-08218-t001:** Simulation parameters.

Sr. No	Parameter	Value
1	Number of Sensors	25
2	Protocol	OpenFlow
3	Controller	Pox
4	Simulation Time Per Iteration	1 min
5	Packets per iteration	10,000
6	Packet Size	512 bytes
7	Bandwidth	10 Mbps
8	Traffic	UDP
9	Shortest Route Calculation	RSIR [46]

**Table 2 sensors-23-08218-t002:** Performance evaluation of the proposed technique and individual classifiers.

Methods	Accuracy	Precision	Recall	F1 Score
**LR**	0.81	0.52	0.47	0.48
**NB**	0.84	0.68	0.73	0.65
**KNN**	0.93	0.82	0.85	0.83
**SVM**	0.94	0.82	0.85	0.83
**RF**	0.99	0.99	0.97	0.98
**DT**	0.99	0.99	0.97	0.98

**Table 3 sensors-23-08218-t003:** Comparative analysis of the data distribution for various parameters.

Features	Min Value	25th Percentile	Median	75th Percentile	Max Value	No of Outliers	IQR	Lower Outlier	Higher Outlier
**Humidity Level**	24	26	28	31	52	106	5	18.5	38.5
**Temperature**	8	18	20	23	32	12	5	18.5	30.5
**Wind Speed**	7	8	9	10	20	98	2	5	13
**Leaf Moisture**	66	84	86	92	97	0	8	72	104
**Soil Temperature**	12	15	17	19	24	0	4	9	25
**Soil Moisture**	6	12	13	14	15	0	2	9	17

## Data Availability

The collection of data used in the research is accessible at the following repository (accessed on 5 August 2023): https://github.com/researchcsaup/IoTs.git.

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
