# Peer review of "AI-Enabled Traffic Control Prioritization in Software-Defined IoT Networks for Smart Agriculture"

_sensors, 2023, doi:10.3390/s23198218_

Round 1

Reviewer 1 Report

The abstract is quite ambiguous. Nothing is clear. What is the motivation? What is the contribution? All are in different directions.

What is the connection between data analyzing and routing? What is the role of SDN?

In the Introduction section, What is the main contribution of the work? Which ML algorithms are used? Why Q learning is used when SDN is automatically managing the routes?

What are the benefits of using the controller in this research?

ML models are used to classify the data. Why these old models are used? What is the novelty of the work?

Are there any enhancements in these models?

You must discuss how these models are working on this data. What layers/modules are working to accomplish this research?

Add some more relevant paper and compare it with a proposed solution.

Ouallane, Asma Ait, Assia Bakali, Ayoub Bahnasse, Said Broumi, and Mohamed Talea. "Fusion of engineering insights and emerging trends: Intelligent urban traffic management system." Information Fusion (2022).

Hussain, Mudassar, Nadir Shah, Rashid Amin, Sultan S. Alshamrani, Aziz Alotaibi, and Syed Mohsan Raza. "Software-defined networking: Categories, analysis, and future directions." Sensors 22, no. 15 (2022): 5551.

Priyadarsini, Madhukrishna, and Padmalochan Bera. "Software defined networking architecture, traffic management, security, and placement: A survey." Computer Networks 192 (2021): 108047.

How Q learning algorithm work here as there are already path-finding mechanisms available in SDN.

There is little information on the implementation details. How ML algorithms are incorporated in SDN controller.

In Figures 6 and 7, several graphs are shown. What is their significance with respect to research output?

More important parameters need to be considered. 

No comparison with state-of-the-art approaches.

English language needs to be improved. 

Reviewer 2 Report

The proposed research addresses a relevant and important challenge in the field of smart agriculture. By combining ML techniques with reinforcement learning (Q-Learning) in the context of SDN, you're poised to contribute to the optimization of data routing, which is vital for the success of smart agricultural systems. This approach has the potential to drive advancements in resource utilization, data processing efficiency, and the overall effectiveness of smart agriculture applications.

1. Provide a detailed explanation of how you plan to integrate ML and Q-Learning to optimize data routing. Specify the types of ML algorithms you intend to use for data classification and how the Q-Learning algorithm will adapt routing decisions based on data classes.

2. Identify gaps in the current research that your proposed ML and Q-Learning approach can address. This will provide context and demonstrate the novelty of your work.

3. Explain the challenges associated with traditional data processing methods for managing the complexity of data in smart agriculture systems?

4. Recent research may be considered to enrich the literature review: An Energy-Efficient and Blockchain-Integrated Software Defined Network for the Industrial Internet of Things and IoT-Driven Artificial Intelligence Technique for Fertilizer Recommendation Model

5. What led you to select the Q-Learning algorithm specifically for optimizing data routing in SDNs? How does it address the dynamic nature of data?

6. Table 3. Box Plot Results, change the table caption and give the meaningful caption.

7. Validate the proposed approach using real-world data or through experimentation in a controlled smart agriculture environment. 

The research outlines a very interesting and promising approach to optimizing data routing in a Software Defined Network (SDN) environment for smart agriculture using a combination of Machine Learning (ML) and the Q-Learning algorithm. This approach seems well-suited to address the challenges of managing the complexity and dynamic nature of data in smart agriculture systems. The paper can be considered after careful revision.

Moderate editing of English language required

Reviewer 3 Report

The authors have proposed an ML-based technique to analyze sensor data from smart agriculture. This technique employs Q-learning for routing and ML algorithms to categorize data into groups. It's interesting to see the application of SDN in agriculture; however, I have the following concerns about the paper:

1. Using ML techniques in SDN is not a novel concept. Why is the specialized technique for "smart agriculture" uniquely devised?

2. Could you provide more details on how the data collection in Section 3.1 was conducted? Information about the types of sensors used, the number of sensors, their placement, the specific types of agriculture studied, and so on would be more convincing.

3. Figure 3 lacks clarity. Architecturally, the SDN system, according to the authors' explanations, would comprise the data, control, and AI planes. Since the ML model categorizes the data plane packets, it seems that the ML models belong to the data plane. Where exactly do the ML models fit in?

4. How were the six ML algorithms chosen for this study? Were other algorithms, such as DNN models or those predicting time-series data, considered?

5. This study assumes the SDN system operates using the OpenFlow protocol. In the real world, SDN systems work with various other SBIs. Can this platform be extended to accommodate them?

6. It appears that the technique merely applies existing ML and Q-learning (RL) algorithms to SDN, a process already seen in numerous other domains. Are there any other modifications or unique features?

7. The title of the paper focuses on control traffic prioritization. Numerous studies exist on control traffic analysis and prioritization, especially using machine learning. Could you compare your work with these and highlight the novelties of your study?

 - "Control Channel Isolation in SDN Virtualization: A Machine Learning Approach," 2023 IEEE/ACM 23rd International Symposium on Cluster, Cloud and Internet Computing (CCGrid), IEEE

 - "Evaluating the SDN control traffic in large ISP networks." 2015 IEEE International Conference on Communications (ICC). IEEE

 - "Comprehensive Prediction Models of Control Traffic for SDN Controllers," 2018 4th IEEE Conference on Network Softwarization and Workshops (NetSoft), IEEE

 - "Scalability of ONOS reactive forwarding applications in ISP networks." Computer Communications 102 (2017): 130-138.

 - "Inter-controller traffic to support consistency in ONOS clusters." IEEE Transactions on Network and Service Management 14.4 (2017): 1018-1031.

NA

Reviewer 4 Report

The introduction could be a little  bit more substantial.

Some other methods used in this area should be listed and compared with the proposed solution, and highlight the advantages of this solution, if any.

The parameters changes in time (Figures 8, 9 and 10) should be more convincingly explained.

If there is a relationships between them, it should be indicated.

The conclusion should be more meaningful and convincing.

Round 2

Reviewer 1 Report

Comments

Some of the previous comments are not properly answered. 

Comment 5 ML models are used to classify the data. Why these old models are used? What is the novelty of the work?

Comment 6 Are there any enhancements in these models?

Comment 7 You must discuss how these models are working on this data. What layers/modules are working to accomplish this research?

Add some more relevant paper and compare it with a proposed solution.

Comment 10 There is little information on the implementation details. How ML algorithms are incorporated in SDN controller.

There is no significant update on the abstract. 

"Traditional data processing techniques in smart agricultural systems face numerous severe hurdles when dealing with the complexities of data management". What are the traditional approaches and what are the challenges? 
The author says "Machine learning and Q-Learning (QL)" Is Q-learning not a machine learning algorithm?

No comparison with existing approaches, i.e., 

Tamizhselvan, C., and V. Vijayalakshmi. "SDN-MCHO: Software Define network based Multi-criterion Hysteresis Optimization based for reliable device routing in Internet of Things for the smart surveillance application." Computer Communications 153 (2020): 632-640.

Sharma, Sudhir, Nitin Chhimwal, Kaushal Kishor Bhatt, Abhay Kumar Sharma, Prashant Mishra, Swati Sinha, Sundeep Raj, and Sandesh Tripathi. "FCS-fuzzy net: cluster head selection and routing-based weed classification in IoT with mapreduce framework." Wireless Networks 27 (2021): 4929-4947.

Naeem, Faisal, Muhammad Tariq, and H. Vincent Poor. "SDN-enabled energy-efficient routing optimization framework for industrial Internet of Things." IEEE Transactions on Industrial Informatics 17, no. 8 (2020): 5660-5667.

Still, there are several English language and grammar mistakes. 

Reviewer 2 Report

The authors have addressed all of my queries. I believe the paper can be accepted for publication.

Author Response

Thank you very much for your kind suggestions.

Reviewer 3 Report

I appreciate the efforts of the authors in comprehending the comments from my previous round.

For the last comment, however, the paper's title is "control traffic prioritization." Although the authors insist that their paper is about data traffic, the title suggests control traffic. Therefore, I suggest the authors at least mention several studies in these fields and explain the differences between them and this study.

NA
